# Selection of memory clinic patients for CSF biomarker assessment can be restricted to a quarter of cases by using computerized decision support, without compromising diagnostic accuracy

**Hanneke F. M. Rhodius-Meester**[1,2]*, **Ingrid S. van Maurik**[1,3], **Juha Koikkalainen**[4], **Antti Tolonen**[5], **Kristian S. Frederiksen**[6], **Steen G. Hasselbalch**[6], **Hilkka Soininen**[7], **Sanna-Kaisa Herukka**[7], **Anne M. Remes**[7,8,9], **Charlotte E. Teunissen**[10], **Frederik Barkhof**[11,12], **Yolande A. L. Pijnenburg**[1], **Philip Scheltens**[1], **Jyrki Lötjönen**[4], **Wiesje M. van der Flier**[1,3]

1 Alzheimer Center Amsterdam, Department of Neurology, Amsterdam Neuroscience, Vrije Universiteit Amsterdam, Amsterdam UMC, Amsterdam, the Netherlands, 2 Department of Internal Medicine, Geriatric Medicine section, Vrije Universiteit Amsterdam, Amsterdam UMC, Amsterdam, the Netherlands, 3 Department of Epidemiology and Biostatistics, Amsterdam Neuroscience, Vrije Universiteit Amsterdam, Amsterdam UMC, Amsterdam, the Netherlands, 4 Combinostics Ltd., Tampere, Finland, 5 VTT Technical Research Centre of Finland Ltd., Tampere, Finland, 6 Department of Neurology, Danish Dementia Research Centre, Rigshospitalet, University of Copenhagen, Copenhagen, Denmark, 7 Department of Neurology, Institute of Clinical Medicine, University of Eastern Finland, Kuopio, Finland, 8 Department of Research Neurology, Unit of Clinical Neuroscience, University of Oulu, Oulu, Finland, 9 MRC Oulu, Oulu University Hospital, Oulu, Finland, 10 Neurochemistry Lab and Biobank, Department of Clinical Chemistry, Amsterdam Neuroscience, Amsterdam UMC, Amsterdam, the Netherlands, 11 Department of Radiology and Nuclear Medicine, Amsterdam Neuroscience, Amsterdam UMC, Amsterdam, the Netherlands, 12 Institutes of Neurology and Healthcare Engineering, UCL, London, England, United Kingdom

* h.rhodius@amsterdamumc.nl

## Abstract

### Introduction

An accurate and timely diagnosis for Alzheimer's disease (AD) is important, both for care and research. The current diagnostic criteria allow the use of CSF biomarkers to provide pathophysiological support for the diagnosis of AD. How these criteria should be operationalized by clinicians is unclear. Tools that guide in selecting patients in which CSF biomarkers have clinical utility are needed. We evaluated computerized decision support to select patients for CSF biomarker determination.

### Methods

We included 535 subjects (139 controls, 286 Alzheimer's disease dementia, 82 frontotemporal dementia and 28 vascular dementia) from three clinical cohorts. Positive (AD like) and negative (normal) CSF biomarker profiles were simulated to estimate whether knowledge of CSF biomarkers would impact (confidence in) diagnosis. We applied these simulated CSF values and combined them with demographic, neuropsychology and MRI data to initiate CSF testing (computerized decision support approach). We compared proportion of CSF

**Data Availability Statement:** The data from this study are available upon request. For this study we requested data from the PredictND consortium. The data-sharing agreement of the PredictND consortium however allows us use of data for this specific project only. The data underlying the results presented in the study are available from the PredictND-board; team@predictnd.eu.

**Funding:** This study is partly funded by Combinostics. The funder provided support in the form of salaries for authors [JK and JL], and had an additional role in the study as Juha Koikkalainen and Jyrki Lötjönen developed the method and quantitative raw data were generated using Combinostics' tools. They also reviewed the manuscript. The specific roles of these authors are articulated in the 'author contributions' section. For development of the PredictAD tool, VTT Technical Research Centre of Finland has received funding from European Union's Seventh Framework Programme for research, technological development and demonstration under grant agreements 601055 (VPH-DARE@IT), 224328 (PredictAD), and 611005 (PredictND). The latter had no role in study design, data collection and analysis, decision to publish, or preparation of the manuscript.'

**Competing interests:** Ingrid M. van Maurik, Antti Tolonen, Kristian S. Frederiksen, Steen G. Hasselbalch, Sanna- Kaisa Herukka, Anne M. Remes and Yolande A.L. Pijnenburg have declared that no competing interests exist. I have read the journal's policy and the authors of this manuscript have the following competing interests: Hanneke F. M. Rhodius- Meester performs contract research for Combinostics, all funding is paid to her institution. Juha Koikkalainen and Jyrki Lötjönen report that Combinostics owns the following IPR related to the paper: 1. J. Koikkalainen and J. Lötjönen. A method for inferring the state of a system, US7,840,510 B2. 2. J. Lötjönen, J. Koikkalainen and J. Mattila. State Inference in a heterogeneous system, US 10,372,786 B2. 3. J. Lötjönen and J. Koikkalainen. Method of inferring a need for medical test, FI 20195603. Koikkalainen and Lötjönen are shareholders in Combinostics. Lötjönen has been an invited speaker for Merck and Sanofi. Hilkka Soininen has served as consultant for ACImmune. Charlotte E. Teunissen has a collaboration contract with ADx Neurosciences, performed contract research or received grants from Probiodrug, Biogen, Esai, Toyama, Janssen prevention center, Boehringer, AxonNeurosciences, Fujirebio, EIP farma, PeopleBio, Roche. Frederik Barkhof receives personal fees from Springer, is a consultant to

measurements and patients diagnosed with sufficient confidence (probability of correct class $\geq 0.80$) based on an algorithm with scenarios without CSF (only neuropsychology, MRI and APOE), CSF according to the appropriate use criteria (AUC) and CSF for all patients.

## Results

The computerized decision support approach recommended CSF testing in 140 (26%) patients, which yielded a diagnosis with sufficient confidence in 379 (71%) of all patients. This approach was more efficient than CSF in none (0% CSF, 308 (58%) diagnosed), CSF selected based on AUC (295 (55%) CSF, 350 (65%) diagnosed) or CSF in all (100% CSF, 348 (65%) diagnosed).

## Conclusions

We used a computerized decision support with simulated CSF results in controls and patients with different types of dementia. This approach can support clinicians in making a balanced decision in ordering additional biomarker testing. Computer-supported prediction restricts CSF testing to only 26% of cases, without compromising diagnostic accuracy.

## Introduction

An accurate and timely clinical diagnosis of Alzheimer's disease (AD) is important both for care and research [1]. Over the last decade the field has moved from a purely clinical concept to a more biological definition of AD [2]. The development of CSF biomarkers has played a pivotal role in this paradigm-shift, enabling accurate detection of AD pathology in vivo [3]. The current diagnostic criteria emphasize the use of these biomarkers in suspected AD [2, 4].

Diagnostic guidelines do not specify which patients should receive CSF testing however, and as a result there is a large practice variation [5]. Recently, a set of criteria for the use of CSF has been published, mirroring the appropriate use criteria (AUC) for amyloid PET, but these are still rather difficult to translate to clinical practice [6]. First, they are phrased in rather general terms, e.g. 'CSF biomarkers can be ordered for all patients with suspected underlying AD pathology', which carries the risk of performing CSF unnecessarily. In addition, the AUC for CSF state that there 'should be a determination of how CSF biomarkers might contribute to the diagnosis and clinical decision making', but it is not self-evident how this should be done. Furthermore, ordering of CSF should also depend on 'the confidence of the clinician in the diagnosis' [6].

Data-driven support tools could be helpful to answer the question 'for which patient should CSF biomarkers be measured?' before embarking on actual testing. By using simulated positive (AD like) or negative (normal) CSF biomarker profiles for a specific patient added to their priori clinical information, such a tool could inform the clinician on how the knowledge of CSF biomarkers would impact (confidence in a) diagnosis [7].

The Disease state index (DSI) classifier is a clinical decision support system (CDSS) which supports the differential diagnosis of several types of dementia by combining and weighing all available information, including cognitive tests, automated MRI features and CSF biomarkers [8]. This multi-class classifier has previously shown to be able to differentiate between several

Bayer, Biogen, Roche, Novartis, Merck, IXICO, GeNeuro and GE Healthcare; receives grants from UK MS Society, g Dutch Foundation MS Research, NWO, NIHR and IMI (H2020), outside the submitted work. Philip Scheltens has served as consultant for Wyeth-Elan, Genentech, Danone and Novartis and received funding for travel from Pfizer, Elan, Janssen and Danone Research. Wiesje M. van der Flier performs contract research for Biogen. Research programs of Wiesje van der Flier have been funded by ZonMW, NWO, EU-FP7, Alzheimer Nederland, CardioVascular Onderzoek Nederland, Gieskes-Strijbis fonds, Pasman stichting, Boehringer Ingelheim, Piramal Neuroimaging, Combinostics, Roche BV, AVID. She has been an invited speaker at Boehringer Ingelheim and Biogen. All funding is paid to her institution. These competing interests do not alter our adherence to PLOS ONE policies on sharing data and materials.

types of dementia [9, 10]. The DSI classifier can achieve high probability of correct diagnosis even without CSF biomarkers, by using an optimal combination of tests [11, 12].

In this work, we took the functionality of our CDSS as an input to predict who would benefit from CSF testing. More specifically, we tested whether the CDSS may help to answer the following question: if a clinician already has information on neuropsychological tests and MRI brain, would additional CSF testing contribute significantly to a more accurate diagnosis? We thus aimed to evaluate how a computer-based tool can help in selecting those patients by using a data-driven approach and simulated CSF outcome.

## Methods

### Subjects

We included 535 subjects from three memory clinics, as part of the PredictND project [13]: 463 subjects from the Amsterdam Dementia Cohort from the Alzheimer center of the Amsterdam UMC (the Netherlands) [14, 15]), 50 subjects from the Danish Dementia Research Centre at Copenhagen University Hospital, Rigshospitalet (Denmark) and 22 subjects from the Neurocenter at Kuopio University Hospital (Finland). The pooled cohort consisted of subjects with the following diagnosis: 286 (53%) AD, 82 (13%) frontotemporal dementia (FTD), 28 (5%) vascular dementia (VaD) and 139 (26%) controls with subjective cognitive decline (SCD). Subjects were eligible for inclusion if both CSF and brain MRI results were available.

All subjects had received a standardized work-up, including medical history, physical, neurological and neuropsychological assessment, MRI, laboratory tests and CSF. Subjects were diagnosed as SCD when the cognitive complaints could not be confirmed by cognitive testing and criteria for mild cognitive impairment (MCI) or dementia were not met. Probable AD was diagnosed using the core clinical criteria of the NIA-AA for AD dementia [4]. Probable FTD (including the behavioural variant of FTD, progressive non-fluent aphasia and semantic dementia) was diagnosed using the criteria from Rasckovsky and Gorno-Tempini, respectively [16, 17]. VaD was diagnosed using the NINDS-AIREN criteria [18]. The local Medical Ethical Committee reviewed and approved the study, including its consent procedure, in accordance with the declaration of Helsinki. Capacity to consent was determined by the clinician, who performed the work-up. All patients provided written informed consent for their data to be used for research purposes.

### Neuropsychology

Cognitive functions were assessed with a brief standardized test battery including widely used tests. We used the Mini-Mental State Examination (MMSE) for global cognitive functioning [19]. For memory we included the Rey auditory verbal learning task (RAVLT) [20]. To measure mental speed and executive functioning we used Trail Making Test A and B (TMT-A, TMT-B) [21]. Language and executive functioning were tested by category fluency (animals) [22]. For behavioral symptoms we used the Neuropsychiatric Inventory (NPI) [23]. Missing data ranged from n = 3 (1%) (MMSE) to n = 126 (24%) (TMT-B).

### Cerebrospinal fluid biomarkers

CSF beta amyloid 1–42 (AB42), total tau and tau phosphorylated at threonine 181 (p-tau) were measured with commercially available ELISA tests (Innotest, Fujirebio, Ghent, Belgium) locally according to standard procedures. Raw data values were used for analysis. Analysis showed comparable values between the centers and hence no further correction was performed.

## APOE genotype

Apolipoprotein-E (APOE) genotyping were performed locally in each center. Patients were dichotomized into APOE e4 carriers (hetero- and homozygous) and non-carriers. APOE data were available in 472 (88%) subjects.

## Imaging markers

MRI images were acquired using 1.5 T or 3 T scanners. The voxel size varied between 0.5–1.1 × 0.5–1.1 × 0.9–2.2 mm for T1 images and 0.4–1.3 × 0.4–1.2 x 0.6–6.5 mm for FLAIR images. We extracted five imaging markers using the cNeuro® cMRI quantification tool: 1) computed medial temporal lobe atrophy (cMTA), 2) computed global cortical atrophy (cGCA), 3) AD similarity scale, 4) Anterior Posterior index and 5) white matter hyperintensities (WMH). First, we automatically computed the MTA and GCA scores [24, 25]. To do so, we defined volumes of brain structures from T1 image segmentations produced by a multi-atlas segmentation algorithm [9, 26]. The cMTA was defined from the volumes of the hippocampi and inferior lateral ventricles separately for left and right. In addition to volumetry, voxel-based morphometry (VBM) [27] was applied to compute gray matter concentrations which were used to define the cGCA. For the AD similarity scale, the region of interest (ROI) in the patient image was represented as a linear combination of the corresponding ROIs from a database of previously diagnosed patients [28, 17]. A ROI-based grading imaging marker was defined as the share of the weights from the linear model having the diagnostic label AD while using the hippocampus area as a ROI. The Anterior Posterior index is a specific measure for characterizing fronto-temporal atrophy, a well-known hallmark in FTD. API measures the relative volume of the cortex in the fronto-temporal and parieto-occipital regions. [29]. In addition, we automatically extracted the volume of WMH from FLAIR images [9, 30]. All imaging markers are corrected for head size [31] and for age and sex [32].

## Disease State Index classifier

The Disease State Index (DSI) classifier is a simple supervised and data-driven machine learning method that compares different diagnostic groups with each other; controls, AD, FTD and VaD in this work [8, 13]. DSI is a continuous value between zero and one, measuring the similarity of all patient data to diagnostic groups defined [8]. For each single test result from cognitive tests, APOE, imaging or CSF, a reference dataset determines a fitness function $f(x) = FN(x)/(FN(x)+FP(x))$ where FN and FP are false negative and positive rates, respectively, when using x as a cut-off to classify patients into two diagnostic groups. Then, this function is used to compute fitness value between zero and one to each test result of the patient. 2) The 'relevance' of each test result is defined as sensitivity+specificity-1 from the reference dataset. 3) DSI is calculated by averaging all fitness values by weighting them with their relevance. This process is repeated for each pairwise comparison (controls-AD, controls-FTD, controls-VaD, AD-FTD, AD-VaD, VaD-FTD). The total DSI for each diagnostic group is the average over all pairwise comparisons, e.g. AD-controls, AD-FTD and AD-VaD for AD. All variables are corrected for age and sex [32]. Due to the design of DSI, there is no need to impute data or exclude cases with incomplete data, as only available data are used [10].

## Probability of correct class

A high total DSI value or a big difference in DSI values between the two most similar diagnostic groups (first and second DSI) provides more confidence for making a diagnosis than a low value or a small difference. Therefore, we define a new measure, probability of correct class

(PCC), which estimates the probability that the suggested diagnosis for a patient is correct, when compared to the clinical diagnosis (ground truth). If the highest DSI value is 0.75 and the second highest is 0.67 for the patient being studied, PCC measures the share of correctly classified cases in the reference database having DSI close to 0.75 and difference close to 0.75–0.67 = 0.08. In other words, PCC estimates (personalized) confidence in classification, i.e., classification accuracy for this very patient. In clinical practice the clinician can adjust the applied PCC cutoff depending on the required accuracy. In this study, patients were considered as having a diagnosis with sufficient accuracy if PCC was at least 0.80. For comparison we performed a sensitivity analysis and repeated our analyses for different PCC cut-offs. Details on the definition and calculation of PCC can be found in the supplement file (S1 File).

## Simulated CSF to guide clinical decision making

In our search for a scenario with the largest proportion of patients diagnosed with high confidence (PCC≥0.80), against the smallest proportion of CSF tested, we applied four scenarios:

- Scenario A ("computerized decision support"): We performed CSF only when predicted to be useful (Fig 1A). This was modelled using a stepwise approach: first, patients diagnosed with PCC≥0.80 based on neuropsychology, MRI and APOE were regarded as clear cases and no CSF was ordered (step one). For the remaining patients with a PCC<0.80, we simulated adding CSF biomarkers; PCC was recomputed after adding positive (age- and sex-normalized median AB42, tau and p-tau values of AD group) and negative (age- and sex-normalized median AB42, tau and p-tau values of SCD group) CSF values (step two) [32]. If either of the resulting simulated PCC values reached the cut-off of≥0.80, actual CSF values were added, and DSI and PCC calculated using these real values (step three).

- Scenario B ("No CSF"): We calculated DSI and PPC for each patient using neuropsychology, MRI and APOE, excluding CSF (Fig 1B).

- Scenario C ("AUC"): We performed CSF based on the AUC criteria (Fig 1C), which state that CSF biomarkers can be applied to all patients with suspected underlying AD pathology [6]. We operationalized this as a DSI for AD >0.6. In all patients with a DSI for AD >0.6, we then calculated PCC adding CSF (step two). For patients with a DSI for AD ≤0.6, no CSF was added and DSI and PCC were calculated using only neuropsychology, MRI and APOE.

- Scenario D ("All CSF"): We performed CSF in all patients (Fig 1D). We calculated DSI and PCC for each patient using neuropsychology, MRI, APOE and CSF.

## Statistical analyses

We used a two-sample test of proportion to test differences in proportion of patients 1) with CSF tested, and 2) diagnosed with sufficient confidence (PCC≥0.80), between the computerized decision support and the other scenarios [34]. We compared the different groups derived from the computerized decision support (scenario A), using Analysis of Variance (ANOVA).

In an additional sensitivity analysis, we visualized the share of patients diagnosed (percentage of patients above PCC cutoff) and the share of patients with CSF measurement for a range of PCC cut-offs (0.5–1.0), and calculated the classification accuracy of the diagnosed patients. This was done for the four diagnostic scenarios: A) the computerized decision support, B) No CSF, C) AUC, and D) All CSF.

Statistical analyses were performed using STATA version 14.1 and R version 3.5.3. A MATLAB toolbox created by [35] was used in the DSI analyses. The analyses were performed in MATLAB version R2015b (MathWorks, Natick, MA, USA).

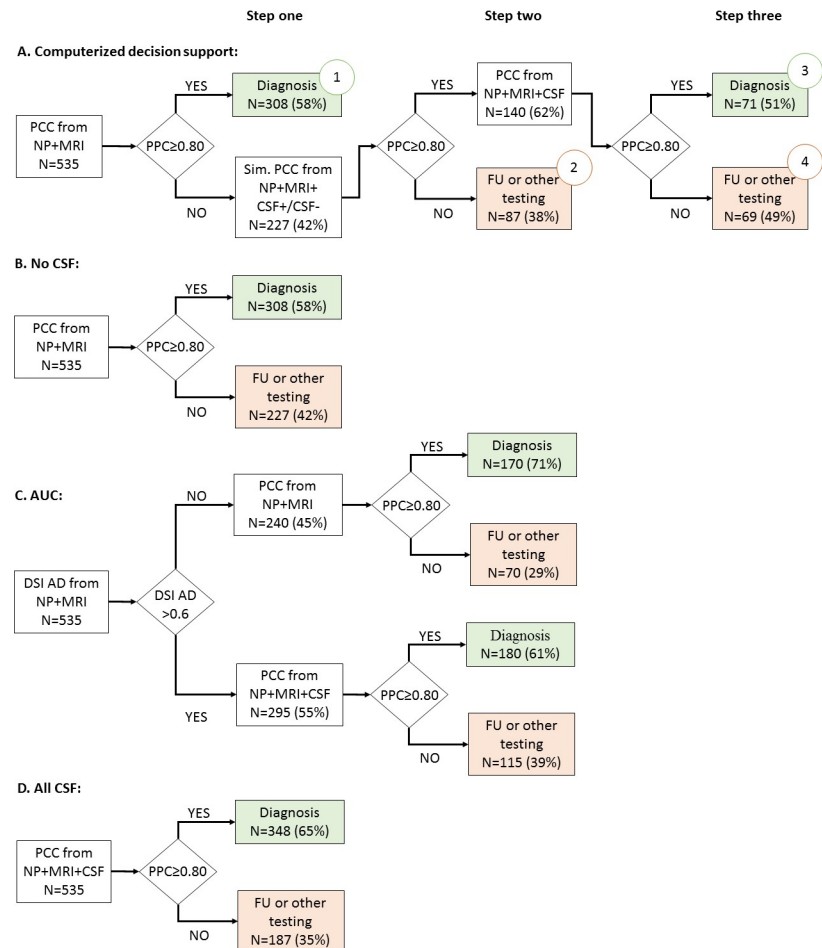

**Fig 1. Flow chart for the four diagnostic scenarios.** AUC: appropriate use criteria according to [33], operationalized as a DSI for AD >0.6, PCC: probability of correct class, NP: neuropsychology, MRI: magnetic resonance imaging, CSF: cerebrospinal fluid biomarkers, Sim.: simulate, FU: follow-up. Numbers in circles denote groups described in section 3.3 and Table 3.

## Results

### Baseline characteristics

Overall mean age was 65±8 and 262 (49%) were females. Details on cognitive tests, MRI markers and CSF results and by group can be found in Table 1.

### Contribution of CSF to diagnosis

To evaluate in which patients the knowledge of CSF changed the confidence in the diagnosis, we examined four diagnostic scenarios, as shown in Fig 1.

In the computerized decision support (Fig 1A) 308 (58%) patients that had PCC≥0.80 in step one (similar to scenario B), did not receive CSF testing. For the remaining 227 (42%) patients, we added simulated CSF values to identify how such values would impact diagnosis. In 140 patients the cutoff of PCC 0.80 was reached (72 with only positive CSF biomarkers, 46 only negative CSF biomarkers and 22 conflicting CSF biomarkers). When we added the actual CSF results of these patients to calculate DSI and PCC (step three), an additional 71 patients reached our threshold to be diagnosed with sufficient confidence. As a result restricting

**Table 1. Baseline characteristics according to baseline diagnosis.**

|  | n = | Control n = 139 | AD n = 286 | FTD n = 82 | VaD n = 28 |
|---|---|---|---|---|---|
| **Female, n (%)** | 535 | 60 (43) | 158 (55) | 36 (44) | 8 (29) |
| **Age, in years** | 535 | 62 ±8 | 67 ±8 | 63 ±6 | 70 ±8 |
| **APOE e4 carrier, n (%)** | 472 | 52 (42) | 165 (65) | 25 (34) | 10 (48) |
| *Neuropsychology* |  |  |  |  |  |
| **MMSE** | 532 | 28 ±1 | 21 ±5 | 24 ±4 | 24±4 |
| **RAVLT, learning** | 506 | 42 ±9 | 22 ±8 | 28 ±8 | 24 ±8 |
| **RAVLT, recall** | 506 | 9 ±3 | 2 ±2 | 4 ±3 | 4 ±3 |
| **TMT-A, in seconds** | 515 | 37 ±20 | 79 ±54 | 57 ±36 | 96 ±57 |
| **TMT-B, in seconds** | 409 | 85 ±39 | 193 ±80 | 155 ±77 | 220 ±86 |
| **Animal fluency** | 519 | 22 ±5 | 13 ±5 | 12 ±7 | 11 ±4 |
| **NPI, total score** | 445 | 7 ±9 | 11 ±11 | 22 ±17 | 16 ±10 |
| *MRI* |  |  |  |  |  |
| **cMTA right** | 535 | 0.3 ±0.5 | 1.4 ±0.8 | 1.6 ±1.2 | 1.5 ±1.0 |
| **cMTA left** | 535 | 0.3 ±0.5 | 1.5 ±0.9 | 2.0 ±1.4 | 1.5 ±1.2 |
| **cGCA** | 535 | 0.3 ±0.5 | 1.2 ±0.8 | 1.3 ±0.8 | 1.4 ±0.7 |
| **WMH, ml** | 535 | 2.9 ±4.7 | 6.6 ±9.9 | 3.6 ±7.7 | 44.5 ±29.6 |
| **AD similarity scale** | 535 | 0.4 ±0.1 | 0.6 ±0.1 | 0.4 ±0.1 | 0.6 ±0.1 |
| **Anterior Posterior index** | 535 | 0.1 ±0.7 | 0.1 ±1.2 | -2.1 ±1.7 | 0.1 ±1.1 |
| *CSF* |  |  |  |  |  |
| **AB42, pg/ml** | 535 | 928 ±280 | 535 ±183 | 914 ±250 | 704 ±252 |
| **Total tau, pg/ml** | 535 | 322 ±208 | 693 ±405 | 337 ±140 | 308 ±162 |
| **P- tau, pg/ml** | 535 | 52 ±24 | 86 ±39 | 45 ±18 | 43 ±18 |
| **AUC+ (DSI for AD>0.6), n (%)** | 535 | 5 (4) | 250 (87) | 31 (38) | 9 (32) |

AD: Alzheimer´s disease, FTD: Frontotemporal dementia, VAD: Vascular dementia, MMSE: Mini-Mental state Examination, RAVLT: Rey Auditory Verbal Learning Test, TMT: Trail Making Test, NPI: Neuropsychiatric Inventory score, cMTA: computed medial temporal lobe atrophy scale (0–4), derived from volume of hippocampus and volume of inferor lateral ventricle, cGCA: computed global cortical atrophy scale (0–3), derived from concentration of cortical grey matter using voxel based morphometry, WMH: volume of white matter hyperintensities, AD similarity scale:based on hippocampus ROI, Anterior posterior index: weighted ratio of volumes of the frontal/temporal lobes and parietal/occipital lobes. MRI volumes are adjusted for head size, AB42: beta amyloid 1–42; p-tau: tau phosphorylated at threonine 181. AUC+: number of patients fulfilling appropriate use criteria according to [33], operationalized as a DSI for AD >0.6.

Data are presented as mean ± SD, unless otherwise specified.

measurement of CSF to only 26% (140/535), diagnosis with sufficient confidence was reached in 71% of the patients ((308+71)/535). In comparison with the other scenarios, the computerized decision support diagnosed a significantly higher proportion of patients diagnosed with sufficient confidence (Table 2). Of note, the computerized approach even outranks scenario D (all CSF), emphasizing that performing CSF in patients were it is not useful, causes confusing and thus lower confidence in the diagnosis. Furthermore, this approach had a significantly lower proportion of CSF tested compared to scenario C based on AUC.

## Groups comparison computerized decision support

When we used the computerized decision support (scenario A, Fig 1A), we can discern different groups. Table 3 shows the baseline characteristics of these different groups. In the largest group 1, patients already had PCC≥0.80 using neuropsychology, MRI and NPO. This group contained mainly controls and patients with a clear clinical profile, i.e. large difference between the highest and second-highest DSI. In group 2, CSF was not helpful as PCC remained <0.80 also after adding simulated CSF values. This group consisted mainly of AD patients, yet with

**Table 2. Comparison of the four diagnostic scenarios.**

| Scenario | Diagnosis with sufficient confidence, PCC≥0.80, n (%) | CSF performed, n (%) |
|---|---|---|
| **A. Computerized decision support** | 379 (71%) [*,†,‡] | 140 (26%)[§] |
| **B. No CSF** | 308 (58%) | 0 (0%) |
| **C. AUC** | 350 (65%) | 295 (55%) |
| **D. All CSF** | 348 (65%) | 535 (100%) |

AUC: appropriate use criteria according to [33], operationalized as a DSI for AD >0.6, PCC: probability of correct class.

Proportion diagnosed

*Scenario A vs Scenario B: difference 13%[7–19], z = 4.44, p<0.001

†Scenario A vs Scenario C: difference 6% [0.4–12], z = 2.104, p = 0.035

‡Scenario A vs Scenario D: difference 6%[0.4–12], z = 2.104, p = 0.035; Proportion CSF performed

§Scenario A vs Scenario C: difference -29% [−35 −−23], z = -9.662, p<0.001

more WMH and a small difference to the second-highest DSI, indicating high probability of comorbid neuropathology. In group 3 CSF was helpful as PCC increased from <0.80 to ≥0.80 by adding simulated CSF, and was ≥0.80 also after adding the actual CSF values (a diagnosis could be made). This group contained mainly FTD and AD patients, where especially tau and p-tau were more abnormal. Finally, in group 4 PCC increased from <0.80 to ≥0.80 by adding simulated CSF, but remained <0.80 when adding the actual CSF values. Patients in group 4 seemed to have multiple underlying neuropathology, with higher tau and more WMH, and a smaller difference with the second-highest DSI.

Nearly all patients in groups 3 and 4 were considered appropriate for CSF following the AUC, showing that our computer-supported decision does not result in a risk of missing patients who would actually have benefited from CSF. In addition, 45% of the patients in group 1 and 30% of the patients in group 2 were considered eligible for CSF testing according to AUC. Yet our computerized decision support predicted that additional CSF testing would not yield additional diagnostic accuracy.

## Different PCC cut-offs

For the current manuscript, we arbitrarily defined an accurate diagnosis as PCC≥0.80. To circumvent this inherent arbitrariness, we repeated our analyses for different PCC cutoffs. Fig 2 shows the share of patients with CSF tested and the share of patients diagnosed for different PCC cutoffs, comparing the four diagnostic scenarios: the computerized decision support, no CSF, AUC and CSF for all patients. Overall, a higher PCC cutoff resulted in a lower number of diagnosed patients. Independent of PCC cutoff, applying no CSF resulted in the smallest share of patients diagnosed, while the stepwise approach gave the largest share of patients diagnosed with CSF tested in max 26.5% patients.

Regardless of PCC cutoff, the accuracy of the four different approaches remained comparable (see S1 Fig) demonstrating that the stepwise approach does not compromise accuracy.

## Visualization of computerized decision support

To enable clinicians to use this method in daily practice, we have visualized the computerized decision support in Fig 3. Using the DSI classifier, clinicians select the PCC threshold (default PCC≥0.80) and simulate positive and negative CSF biomarkers. The resulting change in PCC

**Table 3. Patient groups based on computerized decision support (matching Fig 1A).**

|  | Directly diagnosed | CSF not useful | Diagnosis based on CSF | Not diagnosed | Group wise comparison |
|---|---|---|---|---|---|
|  | Group 1 | Group 2 | Group 3 | Group 4 | P value |
|  | n = 308 | n = 87 | n = 71 | n = 69 |  |
| Female, n (%) | 164 (53) | 31 (36) | 34 (48) | 33 (48) | p = 0.036 |
| Age, in years | 64 ± 8 | 66 ± 8 | 65 ± 8 | 67 ± 8 | p = 0.0045 |
| APOE e4 carrier, n (%) | 150 (55) | 34 (43) | 30 (47) | 38 (66) | p = 0.046 |
| MMSE | 24 ± 5 | 25 ± 4 | 22 ± 4 | 22 ± 5 | p<0.001 |
| *MRI* |  |  |  |  |  |
| cMTA right | 1.1 ± 1.0 | 1.1 ± 1.0 | 1.4 ± 0.8 | 1.4 ± 0.8 | p<0.001 |
| cMTA left | 1.1 ± 1.1 | 1.2 ± 1.1 | 1.5 ± 1.1 | 1.6 ± 1.0 | p = 0.0061 |
| cGCA | 0.9 ± 0.9 | 0.9 ± 0.8 | 1.4 ± 0.8 | 1.3 ± 0.8 | p<0.001 |
| WMH, ml | 7.3 ± 16.5 | 9.1 ± 10.9 | 3.9 ± 4.4 | 7.2 ± 9.8 | p = 0.127 |
| AD similarity scale | 0.5 ± 0.2 | 0.5 ± 0.1 | 0.6 ± 0.1 | 0.6 ± 0.1 | p<0.001 |
| Anterior Posterior index | -0.1 ± 1.5 | -0.4 ± 1.5 | -0.3 ± 1.2 | -0.6 ± 1.3 | p = 0.066 |
| *CSF* |  |  |  |  |  |
| AB42, pg/ml | 737 ± 300 | 696 ± 296 | 653 ± 276 | 619 ± 251 | p = 0.0074 |
| Total tau, pg/ml | 485 ± 365 | 485 ± 315 | 667 ± 399 | 554 ± 380 | p = 0.0018 |
| P- tau, pg/ml | 65 ± 36 | 65 ± 32 | 87 ± 45 | 69 ± 34 | p<0.001 |
| *Clinical diagnosis* |  |  |  |  | p<0.001 |
| Control, n (%) | 109 (78) | 25 (18) | 2 (1) | 3 (2) |  |
| AD, n (%) | 134 (47) | 42 (15) | 55 (19) | 55 (19) |  |
| FTD, n (%) | 44 (54) | 15 (18) | 13 (16) | 10 (12) |  |
| VaD, n (%) | 21 (75) | 5 (18) | 1 (4) | 1 (4) |  |
| Difference with second DSI without CSF | 0.32 ± 0.09 | 0.08 ± 0.05 | 0.11 ± 0.05 | 0.09 ± 0.04 | p<0.001 |
| AUC+ (DSI for AD>0.6), n (%) | 138 (45) | 22 (25) | 67 (94) | 68 (99) | p<0.001 |

AD: Alzheimer´s disease, FTD: Frontotemporal dementia, VAD: Vascular dementia, MMSE: Mini-Mental state Examination, cMTA: computed medial temporal lobe atrophy scale (0–4), derived from volume of hippocampus and volume of inferor lateral ventricle, cGCA: computed global cortical atrophy scale (0–3), derived from concentration of cortical grey matter using voxel based morphometry, WMH: volume of white matter hyperintensities, AD similarity scale:based on hippocampus ROI, Anterior posterior index: weighted ratio of volumes of the frontal/temporal lobes and parietal/occipital lobes. MRI volumes are adjusted for head size. AB42: beta amyloid 1–42; p-tau: tau phosphorylated at threonine 181. Difference with second DSI without CSF: difference between the two most similar diagnostic groups (first and second DSI), AUC+: number of patients fulfilling appropriate use criteria according to [33], operationalized as a DSI for AD >0.6.

Data are presented as mean ± SD, unless otherwise specified

can then support the clinician to refrain from or embark on CSF testing. In Fig 3, two examples of the visualization for clinical use are shown. For patient A, the clinician chose the default PCC cutoff of 0.80. When using only neuropsychology, MRI and APOE data, the highest DSI was obtained for AD with a 72% probability of correct diagnosis. When adding simulated CSF, PCC increased to 0.89 (AD-like CSF), resulting also in a higher DSI for AD. When CSF would be negative, PCC also increased to 0.86 resulting in higher DSI for FTD. As the simulated CSF values resulted in a PCC above the cutoff of 0.80, the suggested advice by the tool is that "CSF measurement is considered potentially useful". Following this advice, the results of the actually performed CSF testing are shown in the bottom panel. The actual CSF biomarker results were AD-like, resulting in a clinical AD diagnosis with 91% probability of correct diagnosis. For patient B, however, simulating only negative CSF increased the PCC above 0.80. And indeed, actual CSF ruled out AD and indicated FTD as the most probable diagnosis. This was also the clinical diagnosis.

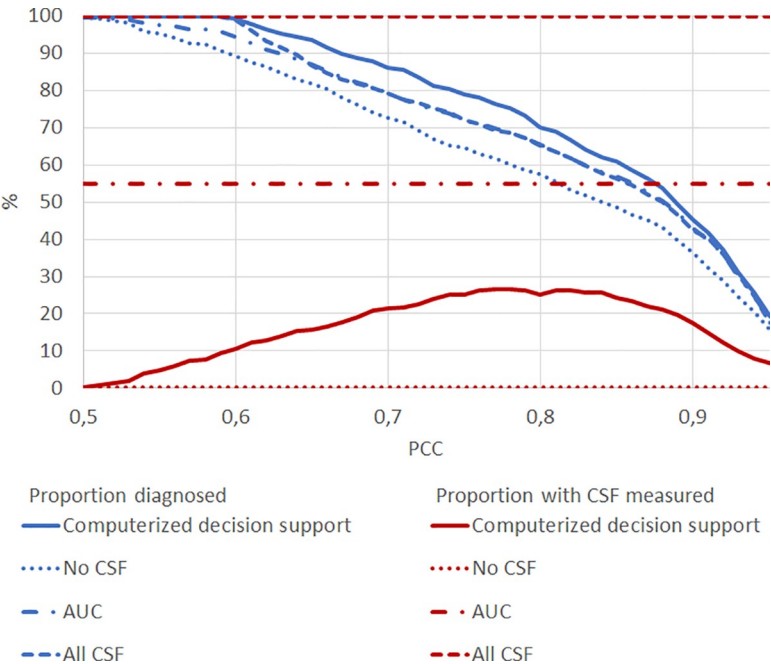

**Fig 2. Share of patients diagnosed and share of patients with CSF measured for different probability of correct class cutoffs, comparing computerized decision support, no CSF, AUC and CSF for all patients.** Blue: proportion of patients diagnosed, Red: proportion of patients with CSF measured, PCC: probability of correct class. Solid lines show results for the computerized decision support (Fig 1A), dotted lines show results for using no CSF, but only neuropsychology, MRI and APOE (Fig 1B), dashed dotted lines show results for AUC (Fig 1C) and dashed lines using all data (Fig 1D).

## Discussion

Although relevant for diagnosing and excluding AD, it remains unclear in which patients applying CSF biomarkers has most added value. This study showed that a data-driven, individualized approach, simulating CSF outcome before embarking on actual CSF testing, can help in identifying those patients in whom CSF biomarkers are most likely to contribute to a more accurate diagnosis. Recommending CSF testing in only one quarter of patients, our computerized decision support led to a diagnosis with sufficient confidence in 71% of the patients. By contrast, in the three other scenarios (CSF in none, based on AUC or CSF in all) less patients were diagnosed with sufficient confidence. By visualizing the effect of simulated CSF (Fig 3) on the probability of correct class (PCC), clinicians can be supported in making a conscious decision in ordering additional biomarker testing.

Neurodegenerative disorders in general and AD in specific, are increasingly considered as biologically defined diseases. Current guidelines and diagnostic criteria therefore emphasize the use of biomarkers [2, 4]. By using these biomarkers, the clinician can either confirm or rule out underlying AD. Yet, clinicians do not often apply CSF biomarkers [36]. The appropriate use criteria (AUC) which have been proposed as a guideline, are generally worded and challenging to translate to clinical practice [6]. In our study, more than half of the patients would be selected for CSF measurement based on fulfillment of the AUC. In addition to specifying criteria, the AUC further mention that the clinician should estimate 'how CSF biomarkers might contribute to the diagnosis and clinical decision making' [6]. Operationalization of this statement is difficult. Tools are therefore needed in search for an answer on the question 'which patient benefits most from CSF?'. To select patients for amyloid measurement, previous

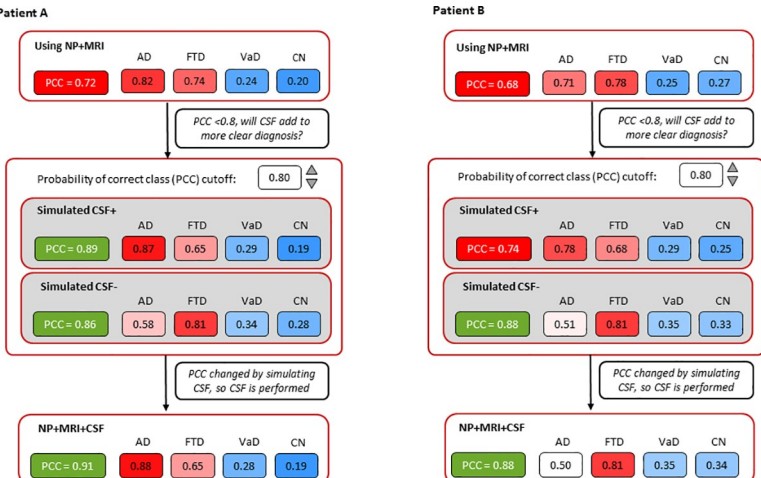

**Fig 3. Examples of visualization simulated CSF for clinical use.** PCC: probability of correct class, AD: Alzheimer´s disease, FTD: Frontotemporal dementia, VAD: Vascular dementia, CN: control. For patient A both simulated positive and negative CSF resulted in an increase of PCC. Adding actual CSF confirmed the AD diagnosis. For patient B simulated negative CSF increased the PCC. Adding actual CSF ruled out AD and indicated FTD as the most probable diagnosis.

studies have aimed to 'predict' amyloid positivity in controls and patients with AD [37–40]. Yet a clinician requires information on how knowledge on amyloid predicts clinical outcome. We added to these studies by using a data-driven approach including not only controls and AD, but also patients with VaD and FTD, and simulated AB42, tau and p-tau values to guide decision making.

In the computerized decision support approach, PCC is first calculated using only neuro-psychology, MRI and APOE data. If the clinician considers the PCC (= the confidence in the correct diagnosis) high enough, no additional testing is needed. This was frequently the case for controls and VaD [11]. When the PCC is too low, simulated normal or AD-like CSF values are added, enabling the clinicians to decide whether the change in PCC is enough for them to order additional testing. If the PCC remains low, the clinician could use this information to refrain from embarking on CSF testing, which would only yield unnecessary costs without improving diagnostic accuracy. Contrariwise the clinician could take other further steps to improve diagnostic certainty, e.g. genetic screening or FDG-PET. One could imagine that comparable approaches would be developed for these other diagnostic tests.

As an example, we used PCC≥0.80 as a cut-off. There is however some inherent arbitrari-ness to the cut-off of diagnostic certainty. It is for example unclear what level of certainty is acceptable in clinical practice. Also, clinicians may value the PCC differently, and may want to use less or more stringent cut-off values. This is why we repeated our analyses for a wide range of cut-offs, resulting in Fig 2. Independent of PCC cut-off, the results remained largely similar: using estimated CSF values to support the choice to embark on testing, helps to reduce costs (smaller proportion CSF testing) while improving diagnostic certainty.

An often mentioned, and important limitation of data-driven decision support systems is that they require multiple pieces of data that are not readily available to the clinician, or that the models are too complex which limit their clinical footprint. In order for clinical decision support tools to be implemented in clinical practice, they should have an understandable basis and be intuitive (i.e. not a complex black box), time efficient, and assist rather than replace the clinician [7]. Our stepwise approach meets all these requirements; well-known CSF values are

simulated in a simple way and the tool provides a suggestion on the usefulness of CSF testing. We used the DSI classifier, an existing, validated machine learning algorithm [10, 12]. The DSI classifier has a graphical counterpart which makes interpretation of results to clinicians more transparent (an example is shown in Fig 3), tolerates missing data (no imputation needed), and gives information about the confidence of the classification.

Furthermore, we used data of three different memory clinic cohorts, warranting a heterogeneous sample. All patients came to the memory clinic seeking medical help; our data thus reflects real-life, clinical patients. In each clinic, patients were diagnosed using a standard clinical workup, and we were able to use measures that overlapped. CSF analyses were performed in three different laboratories, yet we found no significant differences in values between the three centers. MRI scans were acquired on systems with different field strengths, but the automatic analyses of these scans were all performed by the same software [9].

Some limitations also warrant discussion. First, we categorized patients as having single pathologies, while on average 20–40% patients have in fact multiple pathologies, contributing to their syndrome of dementia [41]. This is probably also the case in our sample, as can be seen in diagnostic group 4, described in Fig 1 and Table 3. Here we see a smaller difference with the second DSI (and subsequently a low PCC), suggesting mixed pathology. Second, a limitation could be the use of SCD as controls. However, if SCD is indeed an enriched population, then the ability of our computerized decision support approach to select the correct patient for CSF testing would be underestimated, and would even be better if 'healthy controls' were used. In fact, clinical practice is not about comparing AD patients with healthy controls, but on the differential diagnosis. As SCD patients do seek help for their complaints, they also reflect daily clinical practice, where clinical decision support systems should work.Third, we simulated positive or negative CSF biomarkers and did not include amyloid-PET. This would hamper use of our computerized approach, for example, in the United States, where amyloid-PET is more frequently used than CSF. However, we know CSF correlates strongly with amyloid-PET, especially when both AB42 and tau have been used (as we did here) [42]. Research replicating this study using PET instead, would be a useful addition to this work. Another interesting field for future work is MCI. CSF is a known predictor of progression from MCI to dementia; applying a stepwise approach might be helpful in guiding clinicians also in this challenge [43]. Finally, future studies should evaluate this approach in a prospective design.

The diagnostic routine for AD is very suitable for shared decision making as there is typically more than one reasonable option to choose, such as either perform or not perform additional CSF testing [44]. In shared decision making, the clinician and patient decide together which care plan best fits with individual preferences and needs [45]. An important aspect of shared decision making is that appropriate expectations are set, and recognize that the decision for or against biomarker testing is a highly individualized decision that warrants a shared decision making approach [46]. However, previous research showed that informing patients on diagnostic testing in general was not standard in the memory clinic [36, 44, 47]. The current paper helps clinicians to appreciate whether biomarkers are potentially useful for a specific patient. This information could then be the topic of a dialogue between clinician and patient/ caregiver before actually ordering the test. This in turn aids the clinicians in appropriate expectation setting before embarking on biomarker testing.

In conclusion, we showed that computerized decision support can guide clinicians in who will benefit from CSF and who will not. This resulted in an optimized diagnosis that is at the same time cost-effective. Clinicians can thus order CSF applying a data-driven, individualized approach. By visualizing the effect of simulated CSF on diagnostic certainty, clinicians can be supported in making a weighted decision in ordering additional biomarker testing.

## Supporting information

**S1 File. Probability of correct class (PCC).**
(PDF)

**S1 Fig. Accuracy for different probability of correct class cutoff's, comparing computerized decision support, no CSF, AUC and CSF for all patients.** PCC: probability of correct class Solid lines show results for the computerized decision support (Fig 1A), dotted lines show results for using no CSF, but only neuropsychology, MRI and APOE (Fig 1B), dashed dotted lines show results for AUC (Fig 1C) and dashed lines using all data (Fig 1D).
(TIF)

## Acknowledgments

Research of the Alzheimer center Amsterdam is part of the neurodegeneration research program of Amsterdam Neuroscience. The Alzheimer Center Amsterdam is supported by Stichting Alzheimer Nederland and Stichting VUmc fonds. The clinical database structure was developed with funding from Stichting Dioraphte. Wiesje M van der Flier holds the Pasman chair.

## Author Contributions

**Conceptualization:** Philip Scheltens, Jyrki Lötjönen, Wiesje M. van der Flier.

**Data curation:** Hanneke F. M. Rhodius-Meester, Kristian S. Frederiksen, Steen G. Hasselbalch, Hilkka Soininen, Sanna-Kaisa Herukka, Anne M. Remes, Frederik Barkhof, Yolande A. L. Pijnenburg.

**Formal analysis:** Hanneke F. M. Rhodius-Meester, Ingrid S. van Maurik, Juha Koikkalainen, Charlotte E. Teunissen, Frederik Barkhof, Jyrki Lötjönen.

**Methodology:** Hanneke F. M. Rhodius-Meester, Ingrid S. van Maurik, Juha Koikkalainen, Antti Tolonen, Jyrki Lötjönen, Wiesje M. van der Flier.

**Software:** Juha Koikkalainen, Antti Tolonen, Jyrki Lötjönen.

**Supervision:** Philip Scheltens, Jyrki Lötjönen, Wiesje M. van der Flier.

**Visualization:** Jyrki Lötjönen.

**Writing – original draft:** Hanneke F. M. Rhodius-Meester.

**Writing – review & editing:** Hanneke F. M. Rhodius-Meester, Ingrid S. van Maurik, Juha Koikkalainen, Antti Tolonen, Kristian S. Frederiksen, Steen G. Hasselbalch, Hilkka Soininen, Sanna-Kaisa Herukka, Anne M. Remes, Charlotte E. Teunissen, Frederik Barkhof, Yolande A. L. Pijnenburg, Philip Scheltens, Jyrki Lötjönen, Wiesje M. van der Flier.

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
