## [Decision Letter · Decision Letter 0]

23 Oct 2019

PONE-D-19-25997

Selection of memory clinic patients for CSF biomarker assessment can be restricted to a quarter of cases by using computerized decision support, without compromising diagnostic accuracy

PLOS ONE

Dear Dr. Rhodius-Meester,

Thank you for submitting your manuscript to PLOS ONE. After careful consideration, we feel that it has merit but does not fully meet PLOS ONE’s publication criteria as it currently stands. Therefore, we invite you to submit a revised version of the manuscript that addresses the points raised during the review process.

Please do improve the readability and clarity during the revision to make it easier for general readership. 

We would appreciate receiving your revised manuscript by Dec 07 2019 11:59PM. To enhance the reproducibility of your results, we recommend that if applicable you deposit your laboratory protocols in protocols.io, where a protocol can be assigned its own identifier (DOI) such that it can be cited independently in the future. For instructions see: http://journals.plos.org/plosone/s/submission-guidelines#loc-laboratory-protocols

We look forward to receiving your revised manuscript.

Kind regards,

Han Zhang, Ph.D

Academic Editor

PLOS ONE

Journal Requirements:

2. Please describe in your methods section how capacity to provide consent was determined for the participants in this study. Please also state whether your ethics committee or IRB approved this consent procedure. If it was not necessary to assess the capacity to provide consent of your participants please outline why this was not required in this case

4. Thank you for stating the following in the Financial Disclosure section:

'The author(s) received no specific funding for this work.'

We note that one or more of the authors are employed by a commercial company: Combinostics Ltd and VTT Technical Research Centre of Finland Ltd.

Additional Editor Comments (if provided):

Reviewers' comments:

Reviewer's Responses to Questions

**Comments to the Author**

1. Is the manuscript technically sound, and do the data support the conclusions?

Reviewer #1: Partly

Reviewer #2: Yes

2. Has the statistical analysis been performed appropriately and rigorously? 

Reviewer #1: Yes

Reviewer #2: Yes

3. Have the authors made all data underlying the findings in their manuscript fully available?

Reviewer #1: Yes

Reviewer #2: Yes

4. Is the manuscript presented in an intelligible fashion and written in standard English?

Reviewer #1: Yes

Reviewer #2: Yes

5. Review Comments to the Author

Reviewer #1: The study showed that using a computerized decision support as a tool to guide in selecting patients who need to be detected CSF is helpful in clinical operations. The computerized decision support approach reduced the patients who need CSF testing and classified varied types of dementia with increased sufficient confidence. The work has the potential clinical utility value.

I still have some questions for this work.

1. In this study, I noticed the controls were individuals with subjective cognitive decline. As far as I know, SCD is defined as the population in higher risk for future cognitive impairment and may have already exist abnormal CSF alteration. Thus, I would like to ask why to chose SCD as controls.

2. The description of imaging markers is too general. In this study, T1 images were analysed to extracted quantitative indicators. However, the procedures were not clear. For example,

VBM was conducted by which software and whether to eliminate the effect of volume of cranial cavity. Please specify the processes which would help others to verified the results.

3. In the study, CSF biomarkers included beta amyloid 1-42 (AB42), total tau and tau phosphorylated at threonine 181 (p-tau), which were all AD-related markers. Thus, whether to test these CSF markers would help improve the accuracy of diagnosis in VaD and FTD?

Reviewer #2: 1) Line 131, the reference associated with RAVLT needs to be revised.

2) Line 134, in [21] the paper presents different verbal fluency tests, why the choice of animals ? and did the author try with other categories to see if this leads to different findings ?

3) Line 157, incorrect citation [9], please provide the original references, I suppose the author needs to cite this paper in his/her case; Lötjönen J., Wolz R., Koikkalainen J., Thurfjell L., Waldemar G., Soininen H., Rueckert D. Fast and robust multi-atlas segmentation of brain magnetic resonance images. NeuroImage. 2010;49(3):2352–2365

4) Line 247, in table 1, not all the characteristics are available for all patients, how did the author deal with missing information ? How confident is he/she about their effect on the results.

5) Line 265, it is unclear to me how the author has simulated the CSF values.

6) Line 276, table2 , I would appreciate if the author interprets and explains why scenario A had higher rate than scenario D.

7) According to the figure in the supplementary notes, the higher DSI/delta_DSI the closer PCC to 1. Hence, the higher PCC the better for the diagnosis, how does the author explain that for values larger than 0.8 the accuracy of diagnosis decreased (line 335).

6. PLOS authors have the option to publish the peer review history of their article (what does this mean?). If published, this will include your full peer review and any attached files.

Reviewer #1: No

Reviewer #2: No

---

## [Author Response · Author response to Decision Letter 0]

8 Nov 2019

Amsterdam, 31 October 2019

Regarding PONE-D-19-25997

Dear dr Han Zhang, 

Please find uploaded our revised manuscript for publication in PLOS ONE as part of the special collection on ‘Early diagnosis and treatment of Alzheimer’s disease’. 

We thank the editor and reviewers for their careful reading, thoughtful comments, and recommendation for revision. Please find below our responses to the comments in a point-by-point fashion. We have highlighted changes (via track&change) in response to the reviewers’ comments in the manuscript and supplemental data. 

We hope the rebuttal adequately addresses the points raised during the review process, 

Kind regards, also on behalf of the co-authors, 

Hanneke Rhodius- Meester

Journal Requirements:

Reply: We have updated our manuscript and file naming according to your requirements.

2. Please describe in your methods section how capacity to provide consent was determined for the participants in this study. Please also state whether your ethics committee or IRB approved this consent procedure. If it was not necessary to assess the capacity to provide consent of your participants please outline why this was not required in this case

Reply: We added information on the consent procedure to the paragraph ‘subjects’ in the methods section (L124-126). 

Reply: We address this in our revised cover letter by adding a paragraph on data access. For this study we requested data from the PredictND consortium. The data-sharing agreement of the PredictND consortium however allows us use of data for this specific project only. The data underlying the results presented in the study are available upon request from the PredictND-board; team@predictnd.eu. We hope you understand the restrictions we unfortunately have to deal with.

4. Thank you for stating the following in the Financial Disclosure section:

'The author(s) received no specific funding for this work.'

We note that one or more of the authors are employed by a commercial company: Combinostics Ltd and VTT Technical Research Centre of Finland Ltd.

Reply: We have updated our Funding Statement and Competing Interest Statement, adding that Juha Koikkalainen and Jyrki Lötjönen, as employees of Combinostics, developed the method and quantitative raw data were generated using Combinostics’ tools. They also reviewed the manuscript. The specific roles of these authors are articulated in the ‘author contributions’ section. We confirm that this commercial affiliation does not alter our adherence to all PLOS ONE policies on sharing data and materials and included the following statement: "This does not alter our adherence to PLOS ONE policies on sharing data and materials.” Thank you for changing the online submission form. 

Reply: We updated captions for the supporting information files and in-text citations as requested.

Reviewers' comments:

Reviewer's Responses to Questions

Comments to the Author

1. Is the manuscript technically sound, and do the data support the conclusions?

Reviewer #1: Partly

Reviewer #2: Yes

2. Has the statistical analysis been performed appropriately and rigorously? 

Reviewer #1: Yes

Reviewer #2: Yes

3. Have the authors made all data underlying the findings in their manuscript fully available?

Reviewer #1: Yes

Reviewer #2: Yes

4. Is the manuscript presented in an intelligible fashion and written in standard English?

Reviewer #1: Yes

Reviewer #2: Yes

5. Review Comments to the Author

Reviewer #1: The study showed that using a computerized decision support as a tool to guide in selecting patients who need to be detected CSF is helpful in clinical operations. The computerized decision support approach reduced the patients who need CSF testing and classified varied types of dementia with increased sufficient confidence. The work has the potential clinical utility value.

I still have some questions for this work.

1. In this study, I noticed the controls were individuals with subjective cognitive decline. As far as I know, SCD is defined as the population in higher risk for future cognitive impairment and may have already exist abnormal CSF alteration. Thus, I would like to ask why to choose SCD as controls.

Reply: Thank you for this observation; the use of SCD patients as controls can be seen as a topic of discussion, which we added (p19, L461-467). SCD might be an enriched population, and we think that the reviewers’ concern is that the ability of our computerized decision support approach to select the correct patient for CSF testing would be underestimated. If we used ‘healthy controls’ the tool might even perform better. However, we feel that using SCD patients in this computerized decision support approach is actually a strength of our study. In fact, clinical practice is not about comparing AD patients with healthy controls, but on the differential diagnosis. As SCD patients do seek help for their complaints, they also reflect daily clinical practice, where clinical decision support systems should work. Finally, biomarker status in healthy controls is often unknown, whereas a proportion of these ‘healthy’ controls will be biomarker positive. 

2. The description of imaging markers is too general. In this study, T1 images were analyzed to extracted quantitative indicators. However, the procedures were not clear. For example,

VBM was conducted by which software and whether to eliminate the effect of volume of cranial cavity. Please specify the processes which would help others to verified the results.

Reply: We apologize that the description on imaging markers was unclear to you. As described in the second line of this section, the processes were conducted by the cNeuro® cMRI quantification tool. To enhance readability, and because our core message is not on the imaging markers, we have chosen to show limited details on these markers. These processes of deriving these markers have been described elaborately in several papers, to which we refer in the text. All markers were corrected for age, sex and head size, we added this information to the methods section (L169-170). 

3. In the study, CSF biomarkers included beta amyloid 1-42 (AB42), total tau and tau phosphorylated at threonine 181 (p-tau), which were all AD-related markers. Thus, whether to test these CSF markers would help improve the accuracy of diagnosis in VaD and FTD?

Reply: You are of course correct in stating that the chosen CSF biomarkers are all AD-related markers. In clinical practice these are the CSF biomarkers that clinicians can use for detecting AD pathology during life; CSF biomarkers for FTD and VaD unfortunately have not been developed yet. Nonetheless, CSF biomarkers for AD may still be useful in the differential diagnosis of FTD and VaD. In clinical practice the clinician is faced with a differential diagnostic dilemma’s for all his/her patients. It is thus not about ‘controls versus AD’, but about ‘this type of dementia or that type of dementia’. A tool that supports clinicians in ordering CSF biomarkers in daily practice, should thus be able to cope with this differential diagnostic dilemma. Especially since there can be a broad overlap in clinical symptoms in AD, FTD and VaD. Negative AD-biomarkers increase the likelihood of another underlying pathology, and may in turn help the clinician with this differential diagnostic dilemma. Our computerized decision approach aids for all diagnostic groups, as can be seen in table 3, pointing out both patients in which CSF is not needed and patient in which CSF is useful. For this group with different types of dementia, reflecting clinical practice, we need CSF in only 1 out of 4 patients, without compromising diagnostic accuracy. 

Reviewer #2: 1) Line 131, the reference associated with RAVLT needs to be revised.

Reply: We apologize for this mistake and added the correct reference (L132).

2) Line 134, in [21] the paper presents different verbal fluency tests, why the choice of animals ? and did the author try with other categories to see if this leads to different findings ?

Reply: Thank you for this interesting suggestion. One can definitely argue for example letter fluency, and the difference between letter and animal fluency, can be very indicative. In our dataset (which we used retrospectively) we only had information on animal fluency, not on other categories. However we were able to construct a brief standardized test battery that contains widely used tests, and is thus representative for clinical practice, where our tool should work. 

3) Line 157, incorrect citation [9], please provide the original references, I suppose the author needs to cite this paper in his/her case; Lötjönen J., Wolz R., Koikkalainen J., Thurfjell L., Waldemar G., Soininen H., Rueckert D. Fast and robust multi-atlas segmentation of brain magnetic resonance images. NeuroImage. 2010;49(3):2352–2365

Reply: We added this reference to the revised manuscript (L158). 

4) Line 247, in table 1, not all the characteristics are available for all patients, how did the author deal with missing information ? How confident is he/she about their effect on the results.

Reply: Indeed, information on APOEe4 status and several cognitive tests is missing in a small proportion of the patients. Due to the design of DSI, there is no need to impute data or exclude cases with incomplete data, as only available data are used. Furthermore, we repeated our analysis with whole cases only, and found no relevant change. We added a remark on this to the paragraph on the DSI in the Methods section and included a reference to Tolonen Front Aging Neurosciene 2018 (ref 10), were this is explained in detail (L187-188).

5) Line 265, it is unclear to me how the author has simulated the CSF values.

Reply: We simulated the CSF values by taking age- and sex-normalized median AB42, tau and p-tau values of the AD group (=positive CSF values) and by taking age- and sex-normalized median AB42, tau and p-tau values of the SCD group (=negative CSF values). This age- and sex normalization is done in the same way as for the MRI markers (see ref 31). We described this in the paragraph on ‘Simulated CSF to guide clinical decision making’ in the Methods section and added reference 31 to the text (L216). Hopefully this clarifies the used method.

6) Line 276, table2 , I would appreciate if the author interprets and explains why scenario A had higher rate than scenario D.

Reply: Thank you for the opportunity to elaborate on this interesting finding. Scenario A results in a higher proportion of patients diagnosed with sufficient probability as compared to the other scenario’s (including scenario D), while proportion of performed CSF is much lower. This shows exactly why tools like develop in our study are needed; CSF testing is clearly only useful when it increases the confidence in the diagnosis. One can imagine that borderline CSF values or in case of multiple pathologies, adding CSF values only confuses and leads to lower proportion of diagnosed patients. This can also be seen in f.e. group 2 and group 4 (as shown in Table 3) were CSF did not help. We added a sentence on this to the section above Table 2 (L276-278).

7) According to the figure in the supplementary notes, the higher DSI/delta_DSI the closer PCC to 1. Hence, the higher PCC the better for the diagnosis, how does the author explain that for values larger than 0.8 the accuracy of diagnosis decreased (line 335).

Reply: You are correct in assuming that the PCC correlates to the accuracy of the diagnosis. As can be seen in S1 Fig, the accuracy (as indicated by the black line) increases with increasing PCC, also for values larger than 0.8. Setting the cut-off for a ‘certain’ diagnosis higher, results in less patients diagnosed. This can be seen in Fig 2: the proportion of diagnosed patients decreases with increasing PCC, since fewer patients while have a very high PCC leading to a smaller proportion of patients. Please also note that in this line we refer to the number of patients diagnosed, and not to the accuracy. We hope this explains both Fig 2 and S1 Fig. 

6. PLOS authors have the option to publish the peer review history of their article (what does this mean?). If published, this will include your full peer review and any attached files.

Do you want your identity to be public for this peer review? For information about this choice, including consent withdrawal, please see our Privacy Policy.

Reviewer #1: No

Reviewer #2: No

---

## [Decision Letter · Decision Letter 1]

6 Dec 2019

Selection of memory clinic patients for CSF biomarker assessment can be restricted to a quarter of cases by using computerized decision support, without compromising diagnostic accuracy

PONE-D-19-25997R1

Dear Dr. Rhodius-Meester,

We are pleased to inform you that your manuscript has been judged scientifically suitable for publication and will be formally accepted for publication once it complies with all outstanding technical requirements.

With kind regards,

Han Zhang, Ph.D

Academic Editor

PLOS ONE

Additional Editor Comments (optional):

Reviewers' comments:

Reviewer's Responses to Questions

**Comments to the Author**

1. If the authors have adequately addressed your comments raised in a previous round of review and you feel that this manuscript is now acceptable for publication, you may indicate that here to bypass the “Comments to the Author” section, enter your conflict of interest statement in the “Confidential to Editor” section, and submit your "Accept" recommendation.

Reviewer #1: All comments have been addressed

Reviewer #2: All comments have been addressed

2. Is the manuscript technically sound, and do the data support the conclusions?

Reviewer #1: Yes

Reviewer #2: Yes

3. Has the statistical analysis been performed appropriately and rigorously? 

Reviewer #1: Yes

Reviewer #2: Yes

4. Have the authors made all data underlying the findings in their manuscript fully available?

Reviewer #1: Yes

Reviewer #2: Yes

5. Is the manuscript presented in an intelligible fashion and written in standard English?

Reviewer #1: Yes

Reviewer #2: Yes

6. Review Comments to the Author

Reviewer #1: The manuuscript has been revised thoughtfully.The application of this tool in CSF biomarkers in clinic is valuable. I agree to accept this version.

Reviewer #2: (No Response)

7. PLOS authors have the option to publish the peer review history of their article (what does this mean?). If published, this will include your full peer review and any attached files.

Reviewer #1: No

Reviewer #2: No

---

## [Editor Report · Acceptance letter]

10 Dec 2019

PONE-D-19-25997R1 

Selection of memory clinic patients for CSF biomarker assessment can be restricted to a quarter of cases by using computerized decision support, without compromising diagnostic accuracy 

Dear Dr. Rhodius-Meester:

I am pleased to inform you that your manuscript has been deemed suitable for publication in PLOS ONE. Congratulations! Your manuscript is now with our production department. 

With kind regards,

on behalf of

Dr. Han Zhang 

Academic Editor

PLOS ONE